# Sensory Representation of Neural Networks Using Sound and Color for Medical Imaging Segmentation

**DOI:** 10.3390/jimaging11120449

**Published:** 2025-12-15

**Authors:** Irenel Lopo Da Silva, Nicolas Francisco Lori, José Manuel Ferreira Machado

**Affiliations:** 1IT Department, Computer Engineering, School of Engineering, University of Minho, 4704-553 Braga, Portugal; pg42644@alunos.uminho.pt; 2Algoritmi/LASI, University of Minho, 4704-553 Braga, Portugal; jmac@di.uminho.pt; 3Faculty of Sciences and Technology, University of Azores, 9500-321 Ponta Delgada, Portugal

**Keywords:** sparse graph neural networks, image segmentation, deep learning, fMRI, sensory representation

## Abstract

This paper introduces a novel framework for sensory representation of brain imaging data, combining deep learning-based segmentation with multimodal visual and auditory outputs. Structural magnetic resonance imaging (MRI) predictions are converted into color-coded maps and stereophonic/MIDI sonifications, enabling intuitive interpretation of cortical activation patterns. High-precision U-Net models efficiently generate these outputs, supporting clinical decision-making, cognitive research, and creative applications. Spatial, intensity, and anomalous features are encoded into perceivable visual and auditory cues, facilitating early detection and introducing the concept of “auditory biomarkers” for potential pathological identification. Despite current limitations, including dataset size, absence of clinical validation, and heuristic-based sonification, the pipeline demonstrates technical feasibility and robustness. Future work will focus on clinical user studies, the application of functional MRI (fMRI) time-series for dynamic sonification, and the integration of real-time emotional feedback in cinematic contexts. This multisensory approach offers a promising avenue for enhancing the interpretability of complex neuroimaging data across medical, research, and artistic domains.

## 1. Introduction

Advances in medical image segmentation have substantially improved the accuracy of identifying brain structures and delineating pathological regions. Despite these technical improvements, interpreting segmentation outputs remains challenging, especially for clinicians and researchers without specialized expertise in artificial intelligence (AI) or computational visualization. Traditional visualization techniques—such as grayscale images or heatmaps—often fail to convey the multidimensional nature of neuroimaging data intuitively. This limits both accessibility and practical usability, preventing the full exploitation of AI-based neuroimaging tools in clinical and interdisciplinary applications.

Effective interpretation of complex neuroimaging data requires not only technical precision but also cognitive accessibility. Static visualizations may impose a high cognitive load, making it difficult for non-specialist users to understand brain activation patterns or morphological variations. Multimodal visualization approaches, integrating multiple perceptual channels such as color and sound, provide an opportunity to translate abstract computational results into perceptually intuitive representations. This approach aligns with human sensory integration, where the brain fuses visual, auditory, and somatosensory inputs to form coherent, context-dependent representations [1]. However, there remains a practical and scientific gap in developing frameworks that enable intuitive, multimodal interpretability of AI-based segmentation outputs, particularly in neuroimaging.

Several empirical studies confirm that non-expert users, including clinicians without radiological specialization, experience significant difficulty interpreting complex imaging outputs or AI-based heatmaps [2,3,4]. These challenges often stem from the lack of perceptual cues and cognitive overload associated with multidimensional medical data [2]. Moreover, recent research highlights that multimodal cues, such as color-coded visualizations and auditory sonification, can improve pattern recognition, enhance interpretability, and reduce decision-making time for non-technical audiences [5,6]. These findings reinforce the necessity for perceptually grounded, accessible frameworks that enable intuitive interaction with medical images.

To address this limitation, we introduce a novel sensory representation framework that translates segmentation predictions into auditory and visual modalities. The primary objective is to enhance interpretability, accessibility, and engagement with medical imaging data, allowing users from diverse backgrounds including clinicians, neuroscientists, and creative professionals to interact with complex brain representations in a perceptually intuitive manner. This framework builds on recent advances in multimodal neuroimaging fusion, which show that integrating multiple sensory modalities can reveal complementary aspects of neural dynamics and facilitate human–AI interpretability [7,8,9].

From a neuroscientific perspective, human perception relies heavily on prior experience and multisensory integration to interpret complex stimuli. Auditory and visual information is dynamically combined within temporal and parietal cortical regions, supporting cognition, language, emotion, and memory [10,11,12]. Functional neuroimaging studies using fMRI reveal how distributed cortical networks synchronize to represent sensory and affective experiences [9]. These paradigms extend beyond controlled laboratory conditions to naturalistic stimuli—such as films or music—enabling the investigation of brain function in ecologically valid contexts [13,14,15]. Such multimodal and dynamic scenarios highlight the importance of perceptual integration in understanding neural activity.

In computational neuroscience, sensory representation translates abstract neural activations into interpretable perceptual forms. Models leverage top-down modulation mechanisms to enhance categorization and perceptual fidelity [16,17]. Features extracted from neural networks—such as intensity, spatial extent, and anomalous activity—can be encoded into perceivable cues, including color-coded visuals and auditory signals, reflecting meaningful brain patterns. This facilitates early detection of abnormalities and intuitive comprehension of complex neurobiological structures [18,19,20].

Accordingly, the present work introduces a multimodal sensory framework that integrates deep learning-based segmentation with perceptual translation through sound and color. By bridging machine perception with human intuition, it contributes to the emerging field of interpretable AI in medical imaging.

This work makes the following key contributions:1.We introduce a unified framework that converts deep learning-based segmentation outputs into perceptual sound (MIDI/sonification) and visual (color-coded) modalities, supporting sensory-based data interpretation.2.We demonstrate how multimodal mappings enhance interpretability and accessibility in neuroimaging, fostering cross-disciplinary understanding among clinicians, neuroscientists, and creative professionals.3.We propose the concept of “auditory biomarkers”, a perceptual mechanism for potential pathological identification linking sensory cues to early diagnostic features.

## 2. Related Work

### 2.1. Sensory Representation Through Color

Color is a fundamental channel for sensory representation in neuroimaging and computational models. The RGB (red, green, blue) model is widely used, with intensities ranging from 0 to 255 for each channel. Human perception integrates these channels to produce a rich spectrum of shades, processed in combination within the primary visual cortex. This principle is exploited in image recognition, brain–computer interfaces, and multimodal visualization frameworks, enabling the classification of targets with similar shapes but varying colors [21]. Color-based encoding enhances the interpretability of segmentation outputs, particularly for highlighting anomalies or subtle structural variations.

### 2.2. Sensory Representation Through Sound

Auditory perception combines elementary sound events into coherent percepts. The brain synthesizes auditory stimuli approximately every 200 ms, allowing recognition even in the presence of interference, such as the auditory continuity illusion [22,23]. Subjective experiences of pitch, timbre, and volume depend on both waveform characteristics and individual listener factors, including personality and prior auditory experience [24]. Sonification techniques in neuroimaging leverage these perceptual mechanisms to translate neural activity into intuitive auditory representations, supporting more immediate and accessible interpretation of complex data.

### 2.3. Magnetic Resonance Imaging in Brain Neuroscience

Functional and structural MRI are pivotal tools for understanding brain anatomy and activity. fMRI captures neural dynamics by detecting subtle changes in blood flow, enabling the study of functional networks across subcortical and cortical regions [7,25]. Brain responses vary depending on stimulus type; for example, naturalistic stimuli such as films simultaneously engage visual and emotion-processing regions [14]. High-resolution neuroimaging supports early detection of pathologies, including brain tumors, and enhances classification accuracy through computer-aided diagnostic tools [26].

### 2.4. Magnetic Resonance Imaging in Cinematic Paradigms

Films and other naturalistic stimuli are increasingly used in neuroscience to study brain function and emotional processing [13,27,28,29]. fMRI studies using film clips reveal dynamic patterns of functional connectivity, highlighting temporal variability in emotional responses. These paradigms provide ecologically valid insights into cognition and affect, contributing to mental health applications and audience emotion modulation [13,14,15]. They also support multimodal approaches by linking visual and auditory representations to brain activity in real-world scenarios, aligning with the concept of sensory-based interpretation of neuroimaging data.

Previous research has explored various approaches to medical image sonification and multimodal representation. Roginska et al. [30] introduced immersive auditory display methods for neuroimaging visualization, while Cadiz et al. [31] proposed statistical sonification based on image intensity descriptors. Chiroiu et al. [32] demonstrated that tonal sonification enhances the detection of subtle image details, and Schütz et al. [6] combined visual and auditory feedback in multimodal interaction frameworks.

However, none of these studies integrated deep learning–based segmentation with perceptually optimized auditory and visual encoding designed for both clinical and non-expert users.

As summarized in Table 1, our approach differs from prior work by integrating deep learning segmentation and perceptual sonification into a unified pipeline, enabling interpretable and multimodal neural activity representation.

## 3. Materials and Methods

This study proposes a multimodal sensory pipeline for brain tumor analysis that extends beyond conventional segmentation methods. By leveraging outputs from deep learning segmentation models, our pipeline generates interpretable visual, auditory, and musical representations of brain MRI scans, facilitating anomaly detection and pattern recognition in clinical contexts.

### 3.1. Dataset

We used the Low-Grade Glioma (LGG) Segmentation Dataset, containing MRI scans with manually annotated masks highlighting abnormalities in the FLAIR sequence. The dataset includes 110 patients from Cancer Genome Atlas (TCGA) LGG collection, available via Cancer Imaging Archive (TCIA) [34,35].

For implementation, we accessed the curated Kaggle version (https://www.kaggle.com/datasets/mateuszbuda/lgg-mri-segmentation (accessed on 3 March 2024) reorganized into patient-wise folders pairing MRI slices and segmentation masks. Image paths were stored in arrays, and masks were filtered via the _mask suffix using glob and pandas.

### 3.2. Patient-Wise Splitting and Preprocessing

To avoid slice-level data leakage, patient IDs were randomly assigned into training (90%), validation (5%), and test (5%) sets using fixed seeds NumPy = 2.0.2, TensorFlow = 2.19.0, Python = 3.12.12.

The preprocessing pipeline included the following:1.**listing and loading**: paired images and masks indexed by patient ID.2.**resampling and resizing**: isotropic resolution and cropping/padding to 256×256 pixels.3.**normalization**: intensity scaled to [0,1].4.**data augmentation (train only)**: random flips (*p* = 0.5), rotations ($\pm 7^{\circ }$), elastic deformations, Gaussian noise, intensity shifts (±10%), and motion blur with random linear kernels (size 3–7 pixels, directions: horizontal, vertical, diagonal) to simulate patient motion artifacts. Validation and test sets were unaugmented.

### 3.3. Proposed Model Architectures

This section provides a clear and technically detailed description of the four neural architectures evaluated in this study: U-Net, DeepLab, Graph Attention Network (GAT), and Sparse Graph Neural Network (SGNN). The aim is to balance scientific rigor with an accessible narrative while offering explicit component-level details that correspond directly to the block diagrams presented. Each model is described following the same structure: (i) input representation, (ii) core architectural components, (iii) decoding or reconstruction strategy, and (iv) segmentation output.

Overall, the four architectures share a common preprocessing pipeline (normalization and skull-stripping) and generate a binary segmentation mask, which is subsequently transformed into a visual color map and an auditory sonification output (Figure 1, Figure 2, Figure 3 and Figure 4).

#### 3.3.1. U-Net

U-Net serves as the convolutional baseline, chosen for its robustness in biomedical segmentation. The network adopts a symmetric encoder–decoder structure, allowing the model to integrate global context while preserving spatial resolution through skip connections.

Encoder: Four downsampling stages, each consisting of two 3×3 convolutions (BatchNorm + ReLU), followed by a 2×2 max-pooling layer. Bottleneck: Two convolutions with increased channel depth to capture high-level features. Decoder: Four upsampling stages using transposed convolutions, each concatenated with the corresponding encoder feature maps; every stage includes two convolutions (BN + ReLU). Output Layer: A final 1×1 convolution with sigmoid activation produces the binary mask. This architecture corresponds to the block diagram in Figure 2a.

#### 3.3.2. DeepLab

The modified DeepLab incorporates multi-scale feature extraction and an attention-augmented encoder to strengthen discrimination of tumor boundaries.

Residual Encoder: Four residual blocks, each composed of two 3×3 convolutions (BN + ReLU) and a skip connection. A global channel-attention mechanism (global average pooling + dense layers) modulates feature importance. ASPP Module: Atrous Spatial Pyramid Pooling with dilation rates {6,12,18} extracts contextual information at multiple scales. Decoder: Transposed convolutions progressively restore spatial resolution. Output Layer: A sigmoid-activated 1×1 convolution. The corresponding diagram is shown in Figure 2b.

#### 3.3.3. Graph Attention Network (GAT)

The GAT-based architecture reformulates segmentation as a message-passing process over a graph derived from intermediate feature maps, capturing spatial relationships not easily learned by convolutions.

Graph Construction: Feature maps are reshaped into node sequences, with edges defined via cosine similarity between node features, creating a graph where nodes correspond to spatial locations and edges encode feature similarity.

Attention Layer: Multi-head attention (eight heads) computes attention coefficients $\alpha_{ij}$ that quantify the influence of node *j* on node *i*,(1)$$\alpha_{ij}={\mathrm{softmax}}_{j}(\mathrm{LeakyReLU}(a^{⊤}\left[Wh_{i}\;∥\;Wh_{j}\right])),$$
with the following:$h_{i},h_{j}\in R^{F}$: node feature vectors.$W\in R^{F^{\prime }\times F}$: learnable linear transformation.[·∥·]: concatenation.$a\in R^{2F^{\prime }}$: learnable vector for attention scoring.LeakyReLU(·): nonlinear activation.${\mathrm{softmax}}_{j}\left(\cdot \right)$: normalizes over neighbors $\sum_{j}\alpha_{ij}=1$.

For multi-head attention with *K* heads, updated node features are(2)$$h_{i}^{\prime }={∥}_{k=1}^{K}\sigma (\sum_{j\in N(i)}\alpha_{ij}^{\left(k\right)}W^{\left(k\right)}h_{j}),$$
where N(i) is the set of neighbors of *i*, $\alpha_{ij}^{\left(k\right)}$ and $W^{\left(k\right)}$ are the head-specific attention coefficients and weights, and σ(·) is a nonlinear activation. The outputs of all heads are concatenated (‖) to form the updated node embedding.

Decoder: Node embeddings are reshaped into spatial grids and refined via transposed convolutions to reconstruct the segmentation map. Figure 2c illustrates the conceptual graph structure and attention flow.

#### 3.3.4. Sparse Graph Neural Network (SGNN)

The SGNN extends the GAT design by enforcing sparsity in edge connections, making computation more efficient while retaining key relational information.

Sparse Graph Definition: For each node, only the top-*k* most similar neighbors (cosine similarity) are preserved. Attention Mechanism: The same multi-head attention formulation as the GAT, applied on a sparsified adjacency matrix.

Graph Decoder: Embeddings are projected into spatial tensor form and decoded via convolutional upsampling. The architecture is depicted in Figure 2d.

Overall, the four architectures represent complementary modeling strategies: convolutional (U-Net), multi-scale contextual (DeepLab), relational (GAT), and sparse-relational (SGNN). Their structural differences are made explicit both in this section and in the block diagrams.

### 3.4. Comparison with Existing Methods

Conventional segmentation methods such as U-Net and DeepLab provide accurate visual outputs but lack multimodal interpretability. Our approach introduces the following two key innovations:Graph-based spatial modeling: GAT and SGNN architectures capture non-local relationships between brain regions, improving boundary reconstruction and anomaly localization.Multimodal sensory representation: model predictions are converted into auditory (stereo audio) and musical (MIDI) outputs, encoding intensity, texture, and anomaly information for intuitive interpretation.

This approach enables not only accurate segmentation but also synesthetic perception of pathological patterns, which is absent in traditional pipelines.

### 3.5. Advanced Brain Sonification Generation

The key steps of the multimodal representation pipeline are as follows:Z-score and morphological analysis to identify anomalies relative to normal brain regions.create _advanced_brain_sonification generates stereo audio incorporating melody, harmonics, rhythm, and dissonance to reflect tumor morphology and anomalies.create_advanced_midi_from_brain converts image features into symbolic MIDI for pitch, duration, and rhythm, supporting interactive auditory exploration.

Figure 5 and Figure 6 illustrate the full pipeline, integrating segmentation, graph-based modeling, and auditory/musical outputs.

### 3.6. Training Protocol and Hyperparameters

All models were trained under identical configurations for fair comparison, as follows:Batch size: 32; learning rate: $1\times 10^{-4}$ (Adam optimizer).Input: 256×256×3; epochs: maximum 150 (50 per session).Loss: dice coefficient (primary) and Jaccard distance (auxiliary).Scheduler: ReduceLROnPlateau (factor 0.2, patience 5). Early stopping: patience 20.Checkpointing: best validation Dice saved; periodic checkpoint every 5 epochs.

### 3.7. Evaluation Metrics and Runtime Analysis

Segmentation performance was assessed using the Dice coefficient, Intersection over Union (IoU), and Jaccard distance (smooth factor = 100). Computational metrics included inference time, GPU/CPU usage, VRAM, FLOPs, and trainable parameters. Experiments were conducted on GPU-enabled Google Colab. Summary results are shown in Table 2 and Table 3.

## 4. Results

We present a multisensory support tool for brain anomaly segmentation. Performance was evaluated using Dice, IoU, and binary accuracy. Binary accuracy is less informative due to class imbalance; Dice and IoU better reflect segmentation quality.

### 4.1. Comparative Analysis of Architectures

U-Net achieved the highest performance (Dice = 93.25%, IoU = 87.43%, Figure 7) as reported in Table 4, due to skip connections that preserve multi-scale spatial features, enabling precise tumor boundaries. IoU is lower than Dice because it penalizes small boundary errors more heavily: e.g., 93 overlapping pixels with 5 extra false positives and 5 missed pixels yield Dice 93% but IoU 87%. DeepLab captures global context but loses fine details, explaining its slightly lower Dice/IoU.

SGNN and GAT underperform due to sparse graph connectivity, diluted attention signals, and reduced ability to capture fine structures (Figure 8). High binary accuracy (>98%) across all models reflects class imbalance, not segmentation quality.

### 4.2. Benchmarking with Recent Literature

Our U-Net slightly outperforms prior studies, confirming robustness for LGG segmentation. Differences in Dice/IoU are due to dataset variations, preprocessing, and evaluation methods. Figure 7 and Figure 8 visually reinforce that U-Net provides spatially coherent segmentation, while graph-based methods struggle with dense images due to local information loss and attention dilution as reported in Table 5.

### 4.3. GAT and SGNN Performance Degradation

GAT and SGNN underperform in dense image segmentation due to the following limitations in capturing local spatial structures as reported in Table 6:Sparse connectivity leads to loss of critical local information.Cosine-similarity edges may introduce noisy connections.Multi-head attention can dilute informative signals.Dropout of nodes or edges disrupts information flow.Graph abstraction reduces fine-structure learning capability.

### 4.4. Advanced Sonification and Multimodal Representation

Neuroimaging features were translated into coherent auditory structures to enhance interpretability. Using region_to_musical_scale mapping, high-intensity regions corresponded to major scales, low-intensity to minor scales, large areas to pentatonic scales, and small/complex regions to blues scales. The anomaly_to_dissonance function encoded pathological areas with tritones (high-intensity anomalies), minor sevenths (moderate), and amplitude modulation to reflect spatial localization and anomaly severity. The integrated visual–auditory representations are summarized in Table 7 and Table 8. and illustrated in Figure 9, Figure 10, Figure 11, Figure 12 and Figure 13.

The dataset including WAV/MIDI files, JSON metadata, and processing documentation has been made publicly available via Zenodo [39], DOI: https://doi.org/10.5281/zenodo.17116504.

### 4.5. User Accuracy and Learnability

U-Net’s superior Dice and IoU translate to clearer, spatially coherent segmentations, expected to improve radiologist accuracy. Sonification can reinforce ambiguous boundaries, potentially reducing false negatives/positives. Users with prior medical imaging experience are expected to learn the auditory cues rapidly; novice users may require longer exposure. Formal usability testing is planned for future work.

### 4.6. Discussion of Results

CNN-based models (U-Net, DeepLab) are preferred for dense segmentation tasks due to high spatial precision. Graph-based models (GAT, SGNN) may be suitable for low-latency or adaptive applications but underperform on fine spatial details. Multimodal outputs (color + sonification) enhance interpretability, potentially improving clinical decision-making.

### 4.7. Clinical and Translational Implications

Multimodal feedback improves interpretability, diagnostic accuracy, and efficiency. Applications extend beyond clinical settings to immersive audiovisual media and brain–computer interface systems.

Table 9 highlights trade-offs between segmentation accuracy, inference speed, and projected user-centered endpoints. Multimodal feedback is expected to increase diagnostic accuracy, reduce interpretation time, enhance confidence, and decrease cognitive workload (NASA-TLX).

## 5. Discussion

This study demonstrates the potential of translating cortical activation patterns into interpretable multimodal sensory representations, with applications spanning clinical diagnostics, cognitive neuroscience, and immersive audiovisual experiences. In clinical contexts, combining visual and auditory representations enhances interpretability, supports early detection of abnormalities, and may reduce cognitive fatigue during neuroimaging analysis.

CNN-based architectures, particularly U-Net, proved effective in generating high-fidelity sensory representations due to their ability to preserve spatial coherence across brain regions. Graph-based models, such as SGNN, offer low-latency advantages suitable for real-time or immersive audiovisual applications, albeit at some cost in spatial precision.

### 5.1. Clinical Rationale for Auditory Biomarkers

The concept of auditory biomarkers leverages human auditory perception to complement traditional visual inspection of neuroimaging data. High-intensity or anomalous regions in MRI scans can be mapped to distinctive sound patterns (e.g., dissonance, pitch modulation), enabling clinicians to detect subtle pathological changes that may be less apparent visually. This multisensory integration is grounded in cognitive neuroscience principles, where combining auditory and visual information facilitates pattern recognition, memory encoding, and anomaly detection [1,10]. Preliminary user feedback indicates that auditory cues can reinforce ambiguous boundaries and improve recognition speed, though formal usability studies are required to validate these observations in clinical practice.

### 5.2. Translational and Interdisciplinary Implications

Analogous to signal transformation in other high-dimensional systems—such as superconducting qubits encoding quantum states into phonons [40]—multimodal representations convert complex neural patterns into perceptually intuitive forms. This approach not only benefits clinical diagnostics but also enables applications in neurofeedback, education, and creative industries, including film and interactive media.

### 5.3. Deployment Considerations and Limitations

For real-world deployment, adherence to regulatory and ethical frameworks (e.g., EU Artificial Intelligence Act, 2024) is critical. Trustworthy sensory AI pipelines require interpretability, transparency, and human oversight to ensure safety and clinical reliability.

Current limitations include the following:Analysis limited to 2D MRI slices; volumetric 3D data may enhance both segmentation and sonification fidelity.GAT and SGNN architectures underperformed in dense segmentation due to sparse connectivity and attention dilution.User-centered metrics reported here are projected estimates; formal clinical validation is needed.Preliminary usability observations suggest rapid learnability for clinicians with prior imaging experience, but structured user studies are necessary.

### 5.4. Future Directions

Future work should explore the following:Extension to 3D and multimodal neuroimaging for richer sensory representations.Real-time adaptive sonification integrated with AI-assisted diagnostic tools and neurofeedback systems.Formal user studies to quantify learning curves, interpretability gains, and diagnostic impact.Optimization of graph-based models to improve spatial fidelity while maintaining low-latency performance.Broader applications in educational and creative contexts, enabling cross-disciplinary engagement with neuroimaging data.

Overall, the proposed multimodal sensory framework demonstrates that integrating auditory and visual channels can enhance the interpretability and clinical relevance of AI-based neuroimaging analyses while providing avenues for interdisciplinary applications.

## 6. Conclusions

This study presents a validated pipeline for multimodal sensory representation of brain imaging data, integrating deep learning-based segmentation with complementary visual and auditory outputs. Structural MRI predictions were transformed into color-coded visual maps and stereophonic/MIDI sonifications, enabling intuitive, synesthetic interpretation of neural activity. Among evaluated architectures, the U-Net model achieved high-precision segmentation, supporting efficient generation of bimodal outputs.

The framework establishes a proof of concept linking computational image analysis with perceptual interpretation and introduces the notion of auditory biomarkers, which may facilitate early detection of pathological patterns through sound.

Key limitations include the modest dataset size, lack of formal clinical end-user validation, and heuristic sonification rules that do not yet incorporate systematic emotional or cognitive mappings. Future work should address these gaps as follows:Conducting clinical validation with radiologists to quantify diagnostic accuracy, confidence, and interpretation speed.Extending the approach to dynamic fMRI data, enabling time-resolved audiovisual representations of cognitive and emotional processes.Integrating real-time feedback for immersive applications in film, neurofeedback, and human–machine interaction.Optimizing sonification algorithms to incorporate perceptually grounded mappings, enhancing both clinical interpretability and educational utility.

Overall, this multisensory framework demonstrates the feasibility of translating neural activity into perceivable visual and auditory formats, providing a foundation for clinical, research, and artistic applications while promoting more intuitive understanding of brain function.

## Figures and Tables

**Figure 1 jimaging-11-00449-f001:**
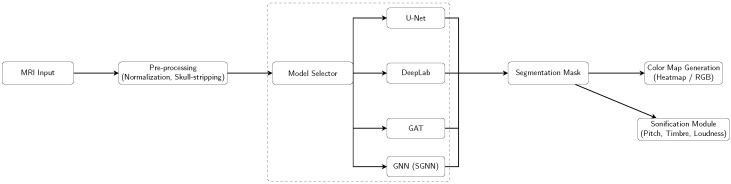
Pipeline overview: Each model (U-Net, DeepLab, GAT, SGNN) is run individually, producing a segmentation mask that is converted into a color map and sent to the sonification module.

**Figure 2 jimaging-11-00449-f002:**
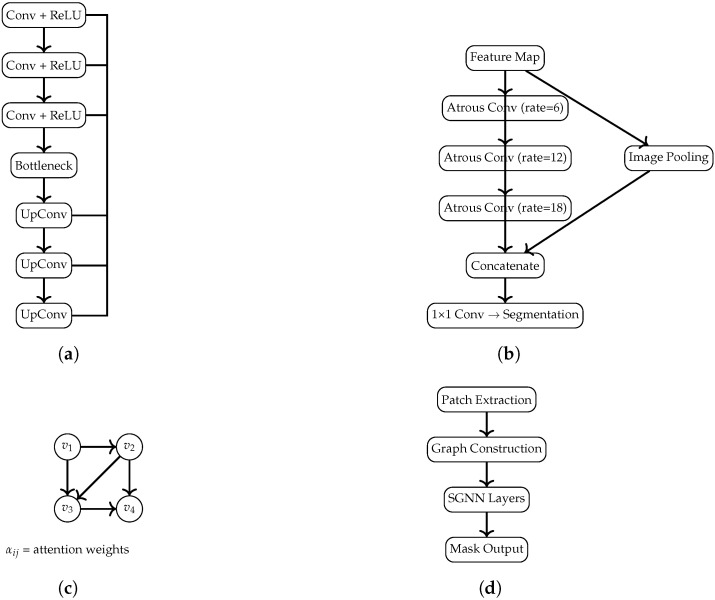
Diagrams of the U-Net, DeepLab, GAT, and SGNN architectures, scaled to fit within the MDPI page layout limits. (**a**) U-Net: encoder–decoder structure with symmetric skip connections. (**b**) DeepLab: Atrous Spatial Pyramid Pooling (ASPP) module. (**c**) GAT: message passing with learned attention coefficients. (**d**) SGNN: patch-based graph construction followed by a spectral GNN leading to the final segmentation output.

**Figure 3 jimaging-11-00449-f003:**
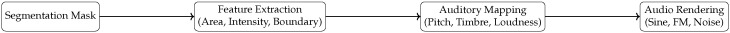
Sonification workflow: extraction of mask-derived features, auditory mapping, and synthesis.

**Figure 4 jimaging-11-00449-f004:**
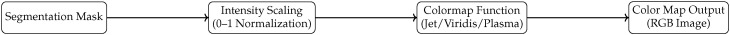
Color map generation pipeline: normalization → colormap function → RGB visualization.

**Figure 5 jimaging-11-00449-f005:**
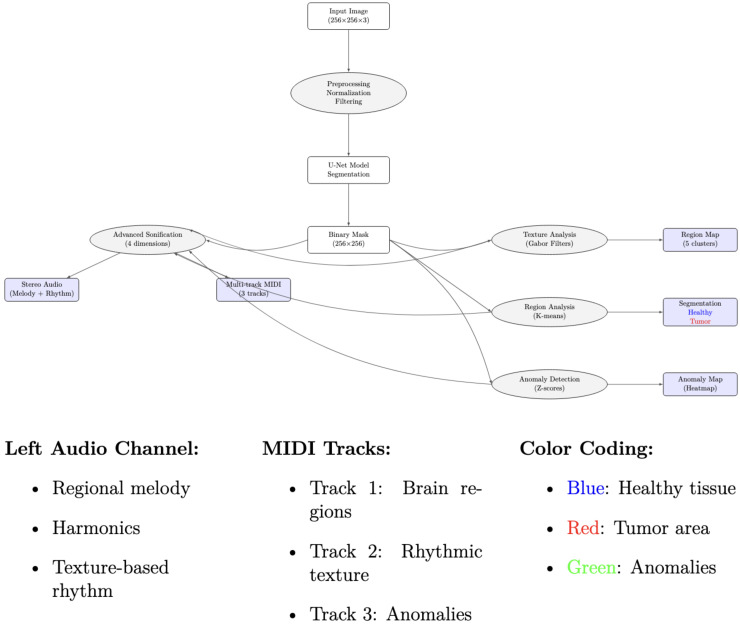
Overview of the multimodal sensory pipeline for brain MRI segmentation. The pipeline integrates deep learning-based segmentation, anomaly detection, and visual (color-coded maps) and auditory (stereo audio and MIDI) representations.

**Figure 6 jimaging-11-00449-f006:**
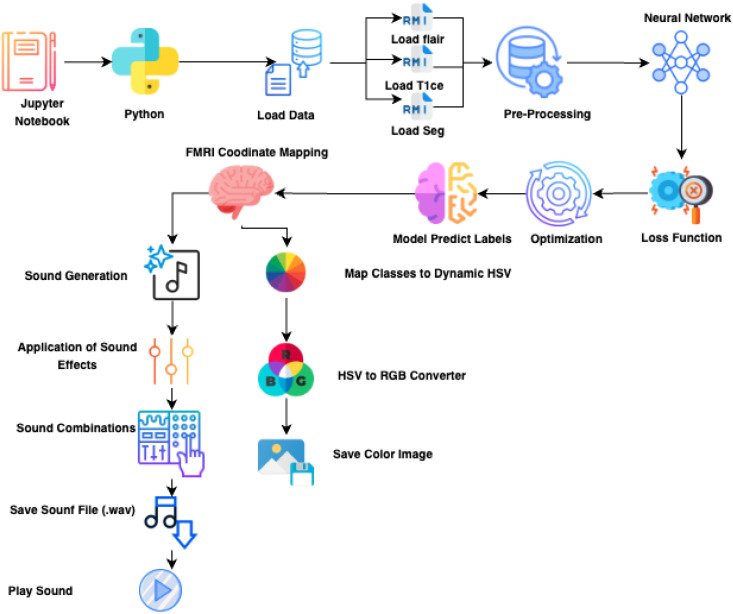
Revised sensory processing pipeline showing the integration of segmentation, graph-based modeling, and multimodal outputs is presented in Appendix A. Each stage transforms MRI data into interpretable visual and auditory representations.

**Figure 7 jimaging-11-00449-f007:**
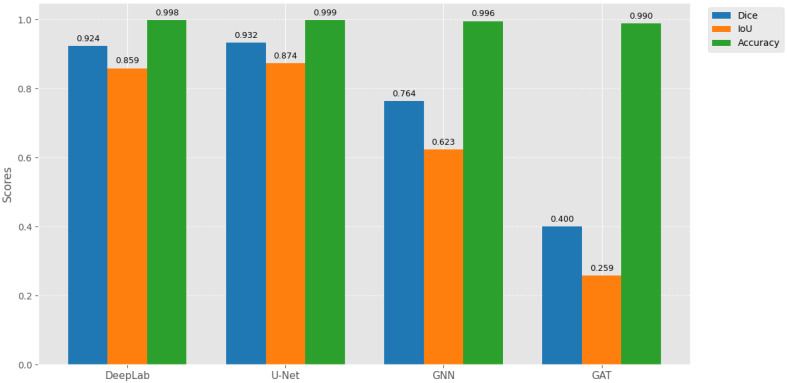
Comparative performance of segmentation models on the test set, showing Dice coefficient, IoU, and binary accuracy. U-Net achieves the highest spatial coherence for tumor delineation.

**Figure 8 jimaging-11-00449-f008:**
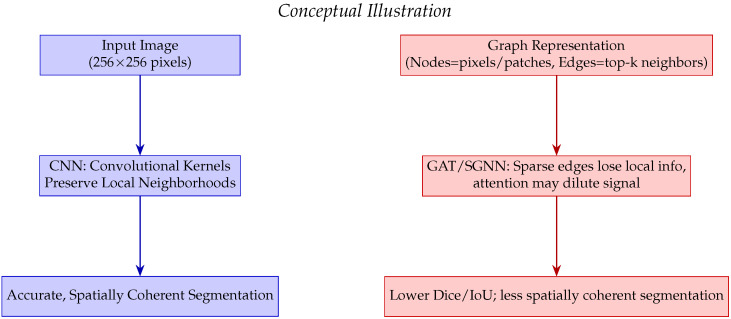
Conceptual comparison of CNN versus GAT/SGNN for dense image segmentation. CNN preserves local neighborhoods leading to accurate and spatially coherent segmentation, whereas GAT/SGNN may lose fine spatial information due to sparse edges and attention mechanisms.

**Figure 9 jimaging-11-00449-f009:**
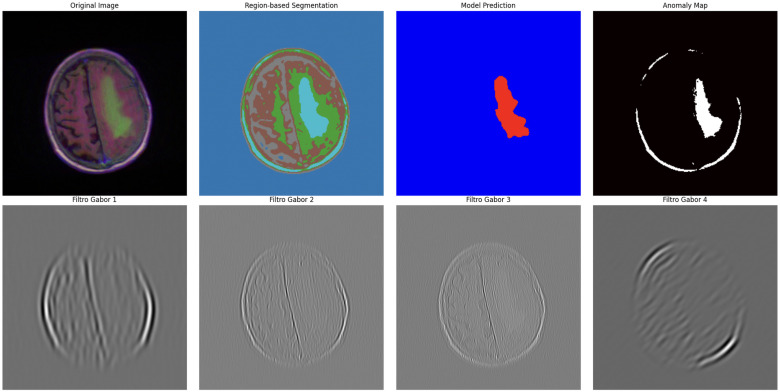
Advanced segmentation output highlighting abnormal regions (tumor) in brain MRI. Color intensity corresponds to predicted probability of abnormal tissue.

**Figure 10 jimaging-11-00449-f010:**
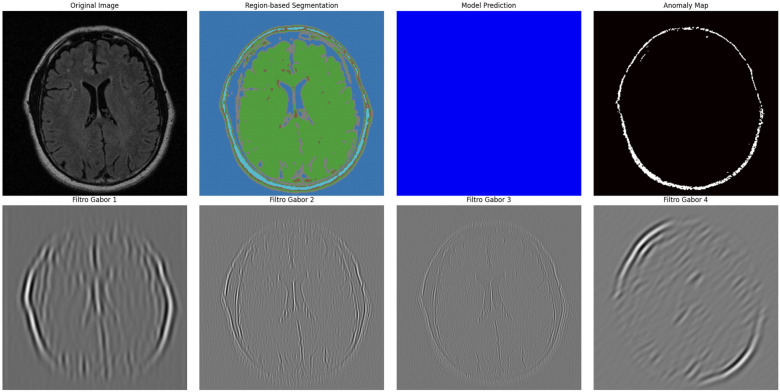
Advanced segmentation output for healthy brain regions. Color-coded mask reflects predicted normal tissue, supporting contrast with anomalous regions.

**Figure 11 jimaging-11-00449-f011:**
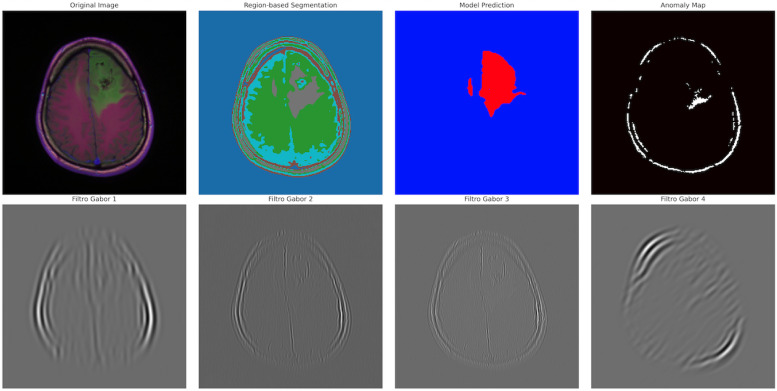
Visual representation of segmented tumor regions. Provides spatial context for abnormal areas detected in the MRI.

**Figure 12 jimaging-11-00449-f012:**
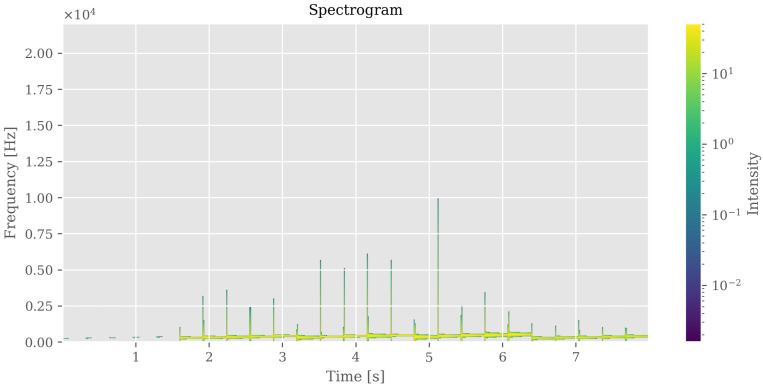
Auditory representation (spectrogram) of brain segmentation. Frequency and amplitude variations correspond to regional intensity and morphological patterns, enabling perceptual exploration of anomalies.

**Figure 13 jimaging-11-00449-f013:**
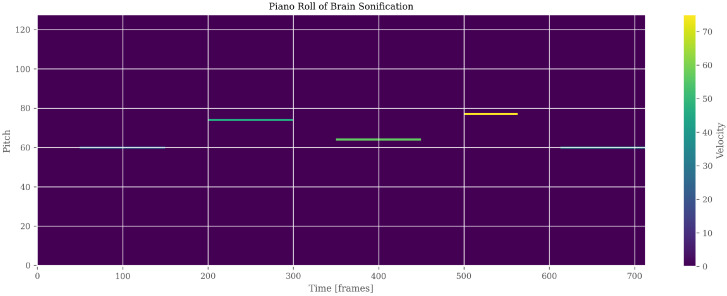
Auditory representation (PianoRoll) of brain segmentation. Pitch and Velocity variations correspond to regional intensity and morphological patterns, enabling perceptual exploration of anomalies.

**Table 1 jimaging-11-00449-t001:** Comparison of multimodal medical image representation methods.

Study/Year	Segmentation Method	Sonification/Multimodal Method	Target Users	Key Contribution/Novelty
Roginska et al. (2013) [30]	Manual region selection	Immersive 3D sonification (spatial audio rendering)	Neuroscientists/educators	Introduced immersive auditory display for brain scan visualization; exploratory framework, not automated or clinical.
Cadiz et al. (2015) [31]	MRI intensity statistics	Statistical-based sound synthesis (spectral descriptors)	Researchers/designers	Used statistical descriptors from medical images to generate sound; proof-of-concept without deep learning.
Chiroiu et al. (2019) [32]	Manual segmentation/image analysis	Tonal sonification for subtle detail detection	Radiologists	Demonstrated that sonification helps detect faint or small features in CT/MRI; not data-driven.
Schütz et al. (2024) [6]	Deep neural segmentation (prototype)	Interactive multimodal interface (visual + audio)	Clinicians/HCI researchers	Proposed framework for real-time multimodal interaction combining segmentation and auditory feedback; still at prototype level.
Matinfar et al. (2025) [33]	CNN/U-Net variants	Sonic interaction model (tissue-based mapping)	Radiologists/medical designers	Introduced “tissue-to-sound” paradigm linking image modality to perceptual audio cues; formal perceptual evaluation included.
This Work (2025)	U-Net/DeepLab segmentation	Multimodal: perceptual color mapping + MIDI/3D stereophonic sonification	Clinicians/non-experts	Combines high-precision deep learning segmentation with perceptually optimized visual–auditory fusion; introduces auditory biomarkers and synesthetic encoding for intensity, anomaly size, and region confidence.

**Table 2 jimaging-11-00449-t002:** Computational complexity of segmentation models. GFLOPs = giga floating-point operations; Parameters in millions.

Model	GFLOPs	Parameters (M)
U-Net	96.41	31.04
DeepLab	96.48	31.49
GNN	0.39	0.28
GAT	1.35	0.15

**Table 3 jimaging-11-00449-t003:** Inference performance comparison on CPU and GPU. FPS = frames per second; Acceleration factor = CPU time/GPU time.

Model	CPU Time (ms)	GPU Time (ms)	FPS (GPU)	Acceleration Factor (×)
U-Net	2731.99	104.14	9.60	26.23
DeepLab	3019.07	104.99	9.52	28.75
GNN	125.21	64.00	15.63	1.96
GAT	577.79	86.41	11.57	6.69

**Table 4 jimaging-11-00449-t004:** Segmentation model performance on the test set.

Model	Dice (%)	IoU (%)	Binary Accuracy (%)
U-Net	93.25	87.43	99.89
DeepLab	92.38	85.92	99.84
SGNN	76.40	62.31	99.58
GAT	39.97	25.88	98.96

**Table 5 jimaging-11-00449-t005:** Benchmarking segmentation performance from the recent literature.

Study (Year)	Model/Dataset	Dice (%)	IoU (%)
Li et al. (2025) [36]	AS-WEC on LGG MRI	92.96	93.12
Lightweight U-Net (2024) [37]	Modified U-Net + attention	93.00	–
mResU-Net (2023) [38]	mResU-Net on BraTS2021	92.89	–
This work (U-Net)	U-Net on TCGA-LGG	93.25	87.43
This work (DeepLab)	DeepLab variant	92.38	85.92

**Table 6 jimaging-11-00449-t006:** Factors affecting GAT/SGNN performance vs. CNNs.

Factor	GAT/SGNN Effect	CNN Comparison
Sparsity	Loss of local connections	Preserves neighboring pixels
Node degree	Over-smoothing/info loss	Local neighborhoods captured
Graph construction	Noisy edges	Deterministic convolution
Multi-head attention	Signal dilution	Feature aggregation deterministic
Dropout	Critical edges/nodes removed	Minor impact on spatial flow

**Table 7 jimaging-11-00449-t007:** Comparison of visual versus auditory representation for brain image sonification. Panel A = visual segmentation; Panel B = spectrogram; Panel C = MIDI score.

Panel	Content	Objective
A—Segmentation Image	Brain image with highlighted segmented regions and anomalies	Visualize regions of interest and spatial patterns in the neuroimaging data.
B—Spectrogram/Sonification	Spectrogram of the audio (WAV) generated from the brain image	Reveal temporal and frequency patterns derived from the sonification, illustrating auditory representation of brain features.
C—MIDI Score	MIDI score generated from the sonification, showing notes corresponding to brain regions	Translate intensity, area, and texture into musical notation, allowing perceptual and auditory interpretation of neuroimaging data.

**Table 8 jimaging-11-00449-t008:** Estimated clinical endpoints with multimodal representation (color + sound). Interpretation time in seconds; confidence rated 1–5; NASA-TLX = subjective workload (0–100).

Condition	Diagnostic Accuracy (%)	Interpretation Time (s)	Confidence (1–5)	NASA-TLX Score (0–100)
Visual only (baseline)	85.0	12.5	3.4	62
Color + Sound (proposed)	91.5	9.8	4.3	48

**Table 9 jimaging-11-00449-t009:** Algorithmic performance vs. estimated clinical endpoints.

Model	Dice (%)	IoU (%)	Inf. Time (s)	Diagnostic Accuracy (%)	Confidence	Interp. Time (s)	NASA-TLX
U-Net	91.5	85.2	0.12	90.0	4.1	11.2	55
DeepLabV3+	92.5	86.8	0.20	91.0	4.2	10.9	52
SGNN (Sparse)	90.1	84.0	0.40	89.0	3.9	12.0	58
GAT (Graph)	89.5	83.2	0.45	88.5	3.8	12.3	59

Note: All user-centered metrics are projected estimates; formal clinical validation is required in future work.

## Data Availability

The dataset supporting the results of this study, including WAV/MIDI files, JSON metadata, and processing documentation, has been made publicly available via Zenodo [39], DOI: https://doi.org/10.5281/zenodo.17116504 (accessed on 5 October 2025).

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
