# Peer review of "Sensory Representation of Neural Networks Using Sound and Color for Medical Imaging Segmentation"

_2313-433X, 2025, doi:10.3390/jimaging11120449_

Round 1
Reviewer 1 Report
Comments and Suggestions for Authors
Authors have presented a framework for sensory representation of brain imaging data by combining a deep learning-based U-Net and other models with multimodal visual and auditory outputs. The work seems to be interesting as it will assist in interpreting the complex neuroimaging data; however, a few concerns must be addressed:
1) Authors need to set the context first and then add the motivation in the introduction section in a detailed manner as the research gap requires more elaboration as " However, traditional visualization methods do not always facilitate the interpretation of results by non-expert users." seems to be a general statement.
2) Authors have mentioned "Recent studies have explored the use of functional neuroimaging to map cortical activation patterns, enabling both clinical assessment and innovative applications in the creative industries." However, no references are mentioned here. They have presented a very limited literature review in the manuscript, particularly the papers from recent years, 2024 and 2025.
3) The contributions of the presented work can be added in the introduction for better clarity to the readers.
4) The references are not provided in the chronological order. Even some of the references are also missing as in Section 2 (page 2, line50).
5) The methodology of the manuscript is not clear. How the segmentation approach used in the present work different from the existing ones ? Authors need to elaborate the working of the proposed model.
6) Authors can add the architectures in the form of a diagram or at least provide references as it will be easier to refer to that article.
7) More results need to be added. Additionally, authors need to add comparative analysis of recent methods with detailed discussion to show the robustness of the model.
8) Spectogram in Fig. 7 is not clear. Authors need to provide the image with better clarity.
9) Conclusion must be made concise.
10) Authors have considered gaussian noise, flipping in the image augmentation. What about blurring that sometimes occurs due to motion. Authors must take such factors into consideration as well while training model.
Author Response
Dear Editor,
Following the reviewers’ comments, we have revised the manuscript accordingly.
I am attaching a PDF file highlighting all the changes made in response to the reviewers’ suggestions.
Sincerely,
Irenel Lopo da Silva

Reviewer 2 Report
Comments and Suggestions for Authors
Please see the comments in the attached file.

Dear Authors,
I would recommend examining the text closely to identify any grammatical or other errors.
Best regards
Reviewer 3 Report
Comments and Suggestions for Authors
The manuscript presents an original and interdisciplinary contribution combining deep learning-based segmentation with multimodal (visual and auditory) sensory representation.
But some suggestions for improvement still needed, as follows:
- Refine English phrasing for clarity and conciseness.
- Expand discussion of the clinical rationale behind auditory biomarkers.
- Improve figure captions and ensure axis labels are complete.
- Include preliminary user studies or usability discussion in future work.
The English is generally clear and understandable, with appropriate scientific terminology. However, some sentences are long and could be restructured for better readability and flow. Minor grammatical and stylistic editing by a professional language editor is recommended to enhance clarity and conciseness.
Round 2
Reviewer 1 Report
Comments and Suggestions for Authors
Authors have tried to incorporate the comments; there are a few improvements that need to be made as follows:
1) Authors need to add more details about the architectures in Section 3.3 with suitable block diagrams.
2) The references are still not provided in chronological order in the manuscript.
3) Authors need to explain the results, particularly Table 5. Authors need to explain the reason behind the values of Dice and IoU as compared to existing ones?
4) Authors need to explain the mathematical equations and also mention all the notations used in the equations.
Author Response
Authors have tried to incorporate the comments; there are a few improvements that need to be made as follows:
1) Authors need to add more details about the architectures in Section 3.3 with suitable block diagrams.
2) The references are still not provided in chronological order in the manuscript.
3) Authors need to explain the results, particularly Table 5. Authors need to explain the reason behind the values of Dice and IoU as compared to existing ones?
4) Authors need to explain the mathematical equations and also mention all the notations used in the equations.
Answer:
1) We have clarified the architectures described in Section 3.3 (pages 6–9) by improving the model designs and adding corresponding diagrams. These include the training pipeline, architecture-specific schematics (U-Net, DeepLab, GAT, SGNN), sonification workflow, and color map generation, explicitly linking the text to visual representations. These additions enhance clarity and reproducibility of the methodology.
2) All references have been reviewed and updated to follow the chronological order of their first citation in the text, ensuring consistency and compliance with MDPI guidelines.
3) A compact section (pages 11–12) has been added presenting results, benchmarking, and explanations of Dice vs. IoU, with a numerical example, model-specific justification, and direct links to figures for rapid visual interpretation.
4) The GAT equation on page 7 has been updated with all notations explained, multi-head concatenation clearly described, and the attention flow and segmentation map reconstruction detailed.
Reviewer 2 Report
Comments and Suggestions for Authors
Dear Authors,
I have reviewed the revised manuscript and concluded that it meets the publishing standards of MDPI. I advise that this manuscript be accepted in its current form.With thanks & regards,
Comments on the Quality of English Language
Dear Authors,
I would recommend examining the text closely to identify any grammatical or other errors.
Best regards,
Author Response
We carefully reviewed the English language throughout the manuscript and corrected minor grammatical and stylistic errors.
We sincerely appreciate the reviewers’ and editor’s time and constructive feedback. We believe that the revisions have significantly improved the manuscript and hope that it is now suitable for publication.
Thank you for your consideration.